# Bidirectional Comorbid Associations between Back Pain and Major Depression in US Adults

**DOI:** 10.3390/ijerph20054217

**Published:** 2023-02-27

**Authors:** Haiou Yang, Eric L. Hurwitz, Jian Li, Katie de Luca, Patricia Tavares, Bart Green, Scott Haldeman

**Affiliations:** 1Center for Occupational and Environmental Health, University of California, Irvine, CA 92617, USA; 2Office of Public Health Studies, Thompson School of Social Work & Public Health, University of Hawaii, Mānoa, Honolulu, HI 96822, USA; 3Department of Environmental Health Sciences, Fielding School of Public Health, University of California Los Angeles, Los Angeles, CA 90095, USA; 4School of Nursing, University of California, Los Angeles, CA 90095, USA; 5Discipline of Chiropractic, School of Health, Medical and Applied Sciences, CQUniversity, Brisbane, QLD 4000, Australia; 6Department of Clinical Education, Canadian Memorial Chiropractic College, Toronto, ON M2H 3J1, Canada; 7Employer Based Integrated Primary Care Health Centers, Stanford Health Care, National University of Health Sciences, San Diego, CA 92121, USA; 8Department of Neurology, University of California, Irvine, CA 92868, USA

**Keywords:** low back pain, depression, comorbidity

## Abstract

Low back pain and depression have been globally recognized as key public health problems and they are considered co-morbid conditions. This study explores both cross-sectional and longitudinal comorbid associations between back pain and major depression in the adult population in the United States. We used data from the Midlife in the United States survey (MIDUS), linking MIDUS II and III with a sample size of 2358. Logistic regression and Poisson regression models were used. The cross-sectional analysis showed significant associations between back pain and major depression. The longitudinal analysis indicated that back pain at baseline was prospectively associated with major depression at follow-up (PR 1.96, CI: 1.41, 2.74), controlling for health behavioral and demographic variables. Major depression at baseline was also prospectively associated with back pain at follow-up (PR 1.48, CI: 1.04, 2.13), controlling for a set of related confounders. These findings of a bidirectional comorbid association fill a gap in the current understanding of these comorbid conditions and could have clinical implications for the management and prevention of both depression and low back pain.

## 1. Introduction

Low back pain and depression have been recognized as major public health problems in the world. Low back pain has been globally ranked the highest cause of disability and years lived with disability among various diseases [1]. Depression has similarly been documented as a leading cause of global health-related burden and disability [2].

Low back pain and depression frequently occur together and are seen as co-morbid conditions [3]. Substantial research has been conducted on the comorbid association between low back pain and depression in the past few decades, but the published research has been marked with inconsistency and controversy [4,5,6,7]. The primary question arising from this literature is whether depression is the cause of low back pain or the result of the chronicity of back pain. There have been three hypotheses related to this question: (a) depression increases the risk of low back pain, (b) low back pain increases the risk of depression, and (c) the association of chronic low back pain and depression is bidirectional [8,9]. Compared with the first two hypotheses, there has been less research investigating the possible bidirectional association hypothesis [6,7,9].

Much of the controversy in the literature can be attributed to the type of study population that has been commonly used, which is that of patients [9]. Patients with low back pain usually have a higher prevalence of depression, and patients with depression have an increased likelihood of symptoms of low back pain than the general population [10].

There have been limited population-based studies and even fewer using U.S. population databases. An initial examination of the cross-sectional association between chronic musculoskeletal pain and depression in the U.S. population indicated significantly increased risk for depression in participants with chronic musculoskeletal pain than those without [11]. However, the data for that study was the first National Health and Nutrition Examination Survey (NHANES I, 1971–1974) and is over a half century old [11]. Other cross-sectional studies conducted in different parts of the world have also indicated a linkage between depression and low back pain [12,13,14].

Longitudinal associations between depression and low back pain have been under studied and the limited evidence has not been consistent [4]. In a randomized controlled clinical trial with 18 months follow-up, Hurwitz and colleagues found bi-directional associations between low back pain and psychological distress using both cross-sectional and longitudinal assessments [6]. Another longitudinal study conducted in Canada focused on a population-based, random sample of adults followed up at 6 and 12 months. This study indicated an independent and robust relationship between depressive symptoms and onset of an episode of spinal pain [4]. However, a third study using adult twins conducted in Spain indicated no significant association between chronic low back pain and the future development of depression [14].

The goal of this study is to explore the cross-sectional and longitudinal comorbid associations between major depression and back pain in a national sample of adults in the U.S. using data from the Midlife in the United States Survey (MIDUS) with a population-based prospective design. The analysis focuses on the comorbid association between depression and back pain, controlling for demographic and socioeconomic factors, and health behavioral factors.

## 2. Materials and Methods

The data used for this study came from the MIDUS, which is aimed at investigating behavioral, psychological, and social factors for health and wellbeing in a national sample of Americans. The MIDUS was developed with a prospective population design. The MIDUS I was conducted in 1995–1996, MIDUS II was conducted in 2004–2006, and MIDUS III was conducted in 2013–2014 [15]. This study used the longitudinal data of MIDUS II and III, with a 9-year follow-up period.

### 2.1. Study Population

The MIDUS collects data through telephone interviews and a self-administered questionnaire (SAQ). In total, 4963 participants who were 30 years of age and above in the MIDUS II were included in the baseline (T-1) for the study, as indicated in Figure 1. However, there were 922 participants who were not part of the SAQ and did not provide data for back pain. An additional 41 participants had no answer to the question on back pain and there were 557 participants with missing data for covariates. Thus, those without data on back pain or covariates were excluded, leaving 3443 participants for T-1, which we used as the sample for the cross-sectional analysis. After 9 years, 882 participants who were lost to follow up, 203 participants who did not have data on back pain (161 participants were not part of the SAQ, and 42 participants had missing data for back pain) at MIDUS III (T-2) were excluded. The final sample size used for the current analysis was 2358 (Figure 1). For the longitudinal analysis on the association between back pain at T-1 and major depression at T-2, we included 2109 participants who were free of major depression at T-1 (249 participant with major depression at T-1 were excluded). For the longitudinal analysis on the association between major depression at T-1 and back pain at T-2, we included 1790 participants who were free of back pain at T-1 (568 participants with back pain at T-1 were excluded).

### 2.2. Measurements

For detailed information on the key variables for the analysis, please see Appendix A.

#### 2.2.1. Back Pain

Back pain was assessed by an independent question that focused on frequency of backaches. A respondent’s answer of experiencing backaches “almost every day” or “several times a week” in the past 30 days was defined as back pain.

#### 2.2.2. Major Depression

Major depression was assessed through a pre-coded dichotomous variable based on the Composite International Diagnostic Interview Short Form (CIDI-SF) [16]. Two domains were included in the assessment: Depressed Affect and Anhedonia. For more information, please see Appendix A.

#### 2.2.3. Health Behaviors

Assessments of health behavioral factors included four variables: leisure-time physical activity, tobacco use, alcohol consumption, and obesity. Leisure-time physical activity was coded as a variable with three categories: active (vigorous or moderate physical activity several times a week), insufficiently active (vigorous or moderate physical activity once a week to less than a month), and inactive (no moderate or vigorous physical activity at all). For more information, please see Appendix A. Current tobacco use was coded as a dichotomized variable with four questions. “Yes” was based on the question, “Do you now smoke cigarettes regularly?”, “No” was based the questions, “Age had first cigarette?”, “Ever smoked cigarettes regularly?”, and “Do you now smoke cigarettes regularly?” Alcohol consumption was coded as a nominal variable with three categories: non-drinkers, light to moderate drinkers, and heavy drinkers. Obesity was assessed based on self-reported weight and height and was classified as a body mass index (BMI) > 30.

#### 2.2.4. Demographic and Socioeconomic Characteristics

Demographic and socioeconomic factors included in the analysis were: sex, age, race/ethnicity, education, and personal earning. Race/ethnicity was coded as two groups: Non-Hispanic White and others. Age was coded into three age groups by years: 30–49; 50–59; and 60–76 and over. Education was assessed through the question: “What is the highest grade of school or year of college you completed?” The response was coded into three categories: high school or less than high school; some college; and college and above. Personal earning was based on the original income variable with a sum of responses to the questions on personal earning income of the respondent, pension income of the respondent, and social security income. It was coded into three categories: <$19,999; $20,000–$59,999; $60,000–$200,000 and above.

### 2.3. Statistical Analysis

The main goal of the analysis is to investigate the question whether major depression is prospectively associated with back pain and whether back pain is linked to subsequent major depression. All data were analyzed using Stata12.1 [17]. Analyses were performed on individuals with complete data.

To describe the characteristics of the study sample at T-1, we first conducted the descriptive analysis on the prevalence of major depression and back pain, characteristics of the study participants in terms of (age, sex, and race/ethnicity), socioeconomic status (education and personal earning), and behavioral factors (leisure-time physical activity, tobacco use, alcohol consumption, and obesity). In addition, we conducted the bi-variate analysis between the two key health outcome variables at T-1 and the characteristics of the participants, with the Pearson’s test.

We then conducted multivariable cross-sectional and longitudinal analyses to explore comorbid associations between major depression and back pain. We constructed models based on several studies that examined the association between back pain and depression, using the demographic characteristics (age, sex, education, and earning) and health behavioral factors (leisure-time physical activity, tobacco use, alcohol consumption, and obesity) as confounders [4,5,6]. Race/ethnicity was not controlled in the four models of cross-sectional and longitudinal associations due to the disproportionally high percentage of Non-Hispanic White participants in the data.

Model 1 focused on cross-sectional comorbid association between major depression at T-1 and back pain at T-1 with multivariable logistic models, controlling for demographic characteristics (age and sex), socioeconomic status (education and personal earning), and behavioral factors (leisure-time physical activity, tobacco use, alcohol consumption, and obesity). Model 2 focused on the cross-sectional association between back pain at T-1 and major depression at T-1, controlling for demographic characteristics, socioeconomic status, and behavioral factors.

Models 3 and 4 were constructed to focus on longitudinal comorbid associations of major depression and back pain with Poisson Regression models. Model 3 focused on longitudinal associations of back pain at T-1 and major depression at T-2, following a group of participants with no major depression at T-1 and controlling for demographic characteristics, socioeconomic status, and behavioral factors. Model 4 focused on major depression at T-1 and back pain at T-2, following a group of participants without back pain from T-1, controlling for demographic characteristics, socioeconomic status, and behavioral factors.

## 3. Results

### 3.1. Baseline Characteristics

Table 1 shows the prevalence of major depression and back pain, characteristics of the study participants, and bi-variate associations at the baseline (T-1). The prevalence of major depression for those with back pain was 17.4%, which was higher than that of the general study population (10.5%) at T-1. The prevalence of back pain within those with major depression (37.4%) was also higher than that of the general study population (22.5%) at T-1.

For demographic characteristics, 55% were female, 36% were aged 60 to 70 and over, and over 90% of the participants were Non-Hispanic White. The bi-variate association between major depression and the main demographic factors were significant, with the exception of race and ethnicity. Age was inversely related to major depression. There was a higher proportion of female participants with major depression (24.1%) and a higher proportion of female participants with back pain (14.1%). For socioeconomic status, education and earning distributions were both inversely related to both major depression and back pain, although the prevalence levels varied. For health behavioral factors, the distribution of the level of leisure-time physical activity was inversely related to complaints of back pain. The lower the level of leisure-time physical activity, the greater the likelihood of back pain. However, current smoking was significantly related to both back pain and major depression.

### 3.2. Cross-Sectional Multivariable Associations

The cross-sectional analysis of multivariable associations is shown in Table 2. Model 1 in Table 2 indicates that major depression at T-1 was significantly associated with back pain at T-1 (aOR 2.13, CI: 1.68, 2.71), controlling for demographic and health behavioral factors. Model 2 in Table 2 shows that back pain at T-1 was significantly associated with major depression at T-1 (aOR 2.11, CI: 1.66, 2.69), controlling for demographic and health behavioral factors. Bidirectional cross-sectional associations between major depression and low back pain were seen.

### 3.3. Longitudinal Associations

In exploring the bidirectional associations between back pain and major depression, Model 3 in Table 3 shows that back pain at T-1 was significantly associated with major depression at T-2 (PR 1.96, t: 1.41, 2.74), controlling for demographic variables and health behavioral factors. Female adults had a prospectively increased risk of major depression (PR 1.87, CI: 1.32, 2.65), and adults who currently smoked at T-1 were more likely to have major depression at T-2 (PR 1.75, CI: 1.17, 2.62). Furthermore, light to moderate drinkers of alcohol at T-1 may have had a lower risk of major depression at T2.

Model 4 in Table 4 shows that major depression at T-1 was associated with back pain at T-2 (PR 1.48, CI: 1.04, 2.12), controlling for a set of confounders. Older adults aged 60 to 75 and older at T-1 were more likely to have back pain at T-2 (PR 1.39, CI: 1.05, 1.84). Furthermore, compared with heavy drinkers, light to moderate drinkers of alcohol at T-1 may have a lower risk of major depression (PR 0.72, CI: 0.53, 0.99).

## 4. Discussion

This study is the first population-based longitudinal study on the bi-directional comorbid association between major depression and back pain in adults in the United States. The findings of this study show that major depression is likely to be prospectively associated with back pain, and that back pain is linked to subsequent major depression. This study provides evidence to support the bidirectional association between these two disabling disorders and is consistent with the findings in the prior study by Hurwitz et al. [7]. This study also readdresses several controversial research issues in terms of hypotheses, study population, measurement, and data analysis in the understanding of the bi-directional associations [3,4,5,6,7,8].

Using data from a national sample of the U.S. population, this study shows the prevalence of major depression and back pain in the U.S. general population. This study shows an increased prevalence of major depression in people reporting back pain (17.4%) when compared to study subjects without back pain (10.5%). At the same time, the prevalence of back pain in study subjects with major depression (37.4%) was higher than those without major depression (8.5%). This finding is consistent with studies conducted in South Korea and Qatar. In the study conducted in South Korea on patients with depressive symptoms, 20.3% reported chronic low back pain, which was much higher than the prevalence of 4.5% of the general population reporting low back pain [13]. The study in Qatar [10] reported a similar pattern, with 13.7% of the general population with depression complaining of low back pain compared to 8.5% in the general population. In this study, male subjects with chronic low back pain reported a higher prevalence of depression compared to the general population. (32% vs. 16%) [18].

One strength of the current study is the instrument used for assessing major depression, the Composite International Diagnostic Interview Short Form (CIDI-SF) [19]. This instrument is considered to have satisfactory reliability and internal consistency [17]. Another strength of the current study was the longitudinal design, which made it possible to explore the impact of major depression as a precursor of back pain compared with a cohort of participants free of major depression, and vice versa.

A main limitation of this study may be attributed to the general goal of the MIDUS, which was not designed for assessing the association between major depression and back pain. The second limitation may be related to the definition of back pain based on the MIDUS question on “backache”. Although it is not clearly defined as conventional “low back pain”, it may imply any spinal pain inferior to the neck, and therefore, could be thoracic pain and could be operationalized as low back pain. The third related limitation is the long follow–up period of 9 years, which was longer than several other published studies using 6- to 12-month follow-up periods [5,7]. With a 9-year follow-up period, different changes in low back pain and depression may be missed. This study sample also had a disproportionately high proportion of Non-Hispanic White racial population, which may limit its generalizability.

Understanding the mechanism of bidirectional association between chronic pain and depression may come from insights provided by recent brain imaging research. Chronic pain and depression appear to have a common neuroplasticity mechanism, which could explain their bidirectional relationship [4,20,21,22]. On the other hand, the bidirectional associations could be explained by shared environmental, clinical, psychosocial, or other factors for back pain and depression [7]. Job strain, a workplace psychosocial factor, has also been linked to both low back pain and major depression [23,24]. However, we did not control for the possible environmental, clinical, or psychosocial factors as confounders. These confounders may be common to both depression and back pain, and they might explain the findings. However, exploring these confounders is beyond the scope of our current study.

This study indicates back pain and depression are not isolated conditions. Understanding the comorbid and bidirectional associations between chronic pain and depression is important, as it may have implications for the management of patients with both depression and low back pain [25,26].

Since both these disorders cause high levels of disability and may be causally related in a bidirectional manner, it would perhaps be of value to assess and manage patients presenting with depression by enquiring about back pain (and vice versa), and addressing those complaints at the same time, rather than considering the management as isolated health concerns. Future population-based longitudinal studies in large scale are needed to explore factors related to the onset, progression, and reoccurrence of low back pain and major depression, as well as psychosocial, behavioral, and other factors that may impact bidirectional comorbid associations.

## 5. Conclusions

This study indicated low back pain and depression are not isolated conditions and that they have a prospective bidirectional association. This study fills a gap in the field and may have implications for the management and prevention of disability associated with both depression and low back pain. Future population-based longitudinal studies in large scale are needed to explore factors related to temporal precedence, onset, progression, and reoccurrence of low back pain and major depression, as well as psychosocial, behavioral, and other factors that may impact bi-directional associations.

## Figures and Tables

**Figure 1 ijerph-20-04217-f001:**
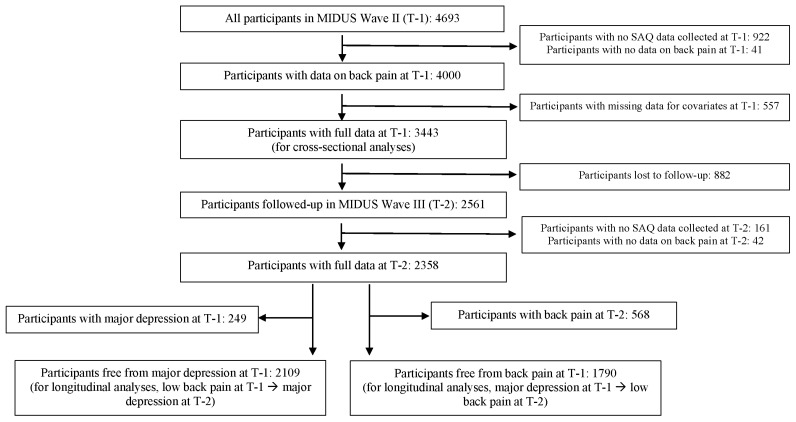
Data flow chart.

**Table 1 ijerph-20-04217-t001:** Prevalence of major depression and back pain at T-1 by characteristics of the study sample at T-1.

		% in General Sample at T1	Back Pain at T-1 (%)	*p*	Major Depression at T-1 (%)	*p*
Health outome						
Major depression at T-1				<0.001		
	No	89.5	20.8			
	Yes	10.5	37.4			
Back pain at T-1						<0.001
	No	77.5			8.5	
	Yes	22.5			17.4	
Demographic characteristic						
Age				0.049		<0.001
	30–49	35.0	20.8		13.6	
	50–59	28.8	21.8		12.1	
	60–75	36.2	24.8		6.2	
Sex				0.005		<0.001
	Male	45.5	20.4		6.1	
	Female	54.5	24.4		14.1	
Race/Ethnicity				0.901		0.731
	Others	8.1	22.8		11.0	
	Non-Hispanic White	91.9	22.5		10.4	
Socioeconomic status						
Education				<0.001		0.01
	High school or less	30.9	27.4		12.3	
	Some college	21.0	23.8		11.5	
	College and more	48.0	18.9		8.9	
Personal earning				<0.001		<0.001
	<$19,999	32.4	29.3		14.1	
	$20,000–$59,999	44.2	20.9		9.8	
	$60,000–$200,000+	23.0	16.3		6.8	
Health behavior						
Leisure-time physical activity				<0.001		0.342
	Active	20.0	17.8		9.0	
	Insufficiently active	37.0	21.4		10.9	
	Inactive	42.0	25.8		10.9	
Tobacco use				<0.001		<0.001
	No	85.0	21.2		8.9	
	Yes	15.0	30.0		19.6	
Alcohol consumption				0.006		0.012
	Non drinkers	40.1	25.3		12.3	
	Light to moderate drinkers	45.1	20.9		8.9	
	Heavy drinkers	14.8	20.0		10.2	
Obesity				<0.001		0.083
	No	68.6	20.4		9.9	
	Yes	31.4	27.1		11.8	

**Table 2 ijerph-20-04217-t002:** Cross-sectional associations with Model 1 and Model 2.

Model 1: Association between Major Depression at T-1 and Back Pain at T-1 *			
	aORs	95% CI	*p*
Major depression at T-1	2.13	(1.68, 2.71)	<0.001
**Model 2: Association between Back Pain at T-1 and Major Depression at T-1 ***			
	**aORs**	**95% CI**	** *p* **
Back pain at T-1	2.11	(1.66, 2.69)	<0.001

* controlling for demographic characteristics (age and sex), socioeconomic status (education and earning) and behavioral factors (leisure-time physical activity, tobacco, alcohol consumption, and obesity) as confounders.

**Table 3 ijerph-20-04217-t003:** Prevalence of major depression at T-2 and longitudinal association between back pain at T-1 and major depression at T-2.

Model 3: Longitudinal Association between Back Pain at T-1 and Major Depression at T-2						
		Major Depression at T-2 (%)	*p*	PR	95% CI	*p*
Health outcome						
Back pain at T-1			<0.001			
	No	22.5		1.00		
	Yes	39.0		1.96	(1.41, 2.74)	<0.001
Demographic characteristic						
Age			<0.001			
	30–49	11.8		1.00		
	50–59	9.9		0.81	(0.56, 1.17)	0.26
	60–75+	5.3		0.55	(0.37, 0.83)	0.01
Sex			<0.001			
	Male	5.6		1.00		
	Female	11.8		1.87	(1.32, 2.65)	<0.001
Socioeconomic status						
Education			0.134			
	High school or less	10.4		1.00		
	Some college	10.0		1.29	(0.79, 2.09)	0.31
	College and more	8.0		0.88	(0.62, 1.26)	0.49
Personal earning			<0.001			
	<$19,999	13.2		1.00		
	$20,000-$59,999	8.1		0.82	(0.56, 1.2)	0.31
	$60,000-$200,000+	5.6		0.84	(0.55, 1.29)	0.43
Health behavior						
Leisure-time physical activity			0.308			
	Inactive	7.7		1.00		
	Insufficiently active	10.0		1.27	(0.89, 1.81)	0.18
	Active	8.9		0.91	(0.56, 1.46)	0.69
Tobacco use			<0.001			
	No	8.0		1.00		
	Yes	16.2		1.75	(1.17, 2.62)	0.01
Alcohol consumption			0.027			
	Heavy drinkers	10.8		1.00		
	Light to moderate drinkers	7.5		0.62	(0.40, 0.99)	0.04
	Non drinkers	9.5		0.75	(0.48, 1.20)	0.23
Obesity			0.009			
	No	8.1		1.00		
	Yes	11.3		1.37	(0.98, 1.91)	0.07

**Table 4 ijerph-20-04217-t004:** Prevalence of back pain at T-2 by characteristics of the study sample at T-1 and longitudinal association between major depression at T-1 and back pain at T-2.

Model 4: Longitudinal Association between Major Depression at T-1 and Back Pain at T-2						
		Back Pain at T-2 (%)	*p*	PR	95% CI	*p*
Health outcome						
Major depression at T-1			<0.001			
	No	7.4		1.00		
	Yes	15.4		1.48	(1.04, 2.12)	0.03
Demographic characteristic						
Age			0.014			
	30–49	21.1		1.00		
	50–59	24.0		1.23	(0.92, 1.63)	0.16
	60–75+	27.3		1.39	(1.05, 1.84)	0.02
Sex			0.019			
	Male	21.8		1.00		
	Female	26.0		1.07	(0.85, 1.34)	0.57
Socioeconomic status						
Education			<0.001			
	High school or less	31.3		1.00		
	Some college	25.1		1.00	(0.69, 1.44)	1.00
	College and more	20.0		0.95	(0.75, 1.21)	0.68
Personal earning			<0.001			
	<$19,999	28.9		1.00		
	$20,000–$59,999	24.5		0.96	(0.74, 1.26)	0.79
	$60,000–$200,000+	17.6		1.07	(0.80, 1.43)	0.64
Health behavior						
Leisure-time physical activity			<0.001			
	Inactive	19.3		1.00		
	Insufficiently active	21.4		0.79	(0.61, 1.02)	0.08
	Active	29.3		0.82	(0.61, 1.11)	0.21
Tobacco use			0.002			
	No	23.1		1.00		
	Yes	31.0		1.03	(0.74, 1.45)	0.85
Alcohol consumption			0.004			
	Heavy drinkers	26.6		1.00		
	Light to moderate drinkers	20.9		0.72	(0.53, 0.99)	0.05
	Non drinkers	27.4		0.82	(0.59, 1.13)	0.23
Obesity			0.002			
	No	22.3		1.00		
	Yes	28.3		1.23	(0.97, 1.57)	0.09

## Data Availability

The datasets analyzed during this study are available in ICPSR, the Inter-university Consortium for Political and Social Research, https://www.icpsr.umich.edu/web/ICPSR/search/studies?q=MIDUS (accessed on 2 January 2021).

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
