# Peer review of "Bidirectional Comorbid Associations between Back Pain and Major Depression in US Adults"

_ijerph, 2023, doi:10.3390/ijerph20054217_

Round 1
Reviewer 1 Report
I appreciate the invitation to review this manuscript, which I find very appropriate and of interest for its publication. I only have two questions and/or proposals for the authors:
In the results section, it seems difficult to understand the distribution of the tables, it is not identified if it is table one that is divided into several sections by model, since I cannot find a footer or description of the table, and between each model there is a paragraph with result description, so is it the same table or are they different tables?
How can you ensure that low back pain is due to major depression and not due to age itself or some age-associated comorbidity?

Reviewer 2 Report
Thank you for giving me the possibility to review this article. It is very interesting the objective of this work which is to explore the cross-sectional and longitudinal associations between major depression and low back pain in a national sample of adults in the US using data from the Midlife in the United States Survey (MIDUS) with a population-based prospective design.
It is not so much the changes but the doubts I have about their results and how they are explained.
Figure 1 is illegible, it should be possible to read it correctly.
In point 2.2.1 Low back pain
Low back pain was evaluated by means of a separate question focused on the frequency of back pain. Low back pain was defined as the respondent's response of experiencing back pain "almost every day" or "several times a week" in the last 30 days.
The question they ask if you have back pain is very general, they should ask if you have back pain referred to the lower back. If not, they are not talking about low back pain, but about back pain without specifying a specific location.
Although they consider it as a limitation, I would like to point out the very long period from one questionnaire to another. The results should be taken with certain uncertainties regarding the associations, since in the variable back pain, the question only considers having back pain in the last 30 days. To consider that there is a bidirectional association is very courageous on their part.
It is interesting in terms of their approach to the control of the subject, whether they have depression or back pain, although it would not be superfluous to include in the manuscript (since the objective of the MIDUS questionnaires was to investigate behavioral, psychological and social factors of health and well-being in a national sample in the United States), that variables be studied with questions that reflect their physical and cognitive situation during the last year.
You should review the bibliography if you follow the Vancouver format.
For example number 3, modify:
Bair, M. J.; Robinson, R. L.; Katon, W.; Kroenke, K. Depression and pain comorbidity: a literature review. Arch Intern Med 2003, 163, (20), 2433-45.
By: Bair MJ, Robinson RL, Katon W, Kroenke K. Depression and pain comorbidity: a review of the literature. Arch Intern Med. 2003;163(20):2433-45. doi: 10.1001/archinte.163.20.2433.
Reviewer 3 Report
General questions
Congratulations for the nice work. This work presents a good scientific relevance. However, some issues need to be clarified in addition to other small corrections.
Major questions
- I suggest changing keyword and to use different words title. This increases the chances of searching for the manuscript.
- Line 171: “distribution of level of leisure time physical activity was positively related to low back pain...”: this is not what table 1 shows. In the table 1, the lower the level of leisure time physical activity, the greater the back pain (ie the relationship is negative)
- it was not described in the text results of table 1 referring to Alcohol consumption
- Table 1 and Table 2: From Education the data are not organized (titles vs subtitles: first column) = (Education; Personal earning; Leisure time Physical activity; Tobacco use; Alcohol consumption)
- Table 1: unconfigured in some parts
- Line 290: “The guidelines recommend adults do 150 to 300 minutes of moderate-intensity physical activity or 75 to 150 minutes of vigorous-intensity physical activity.[27]”: Please, by week or day?
Minor questions
- Line 177: 3.2. . Cross-sectional multivariable associations (extra dot before title)
- Table 2: Model 2: Association between low back painat T-1 and major depression at T-1* (painat)
- Line 195: Prevalence f major depression
- Line 195 and Line 196: T-2 and T2 (?) (other lines too: 194, 202...)
- Line 201: T-1may
- Line 203: ...longitudinal association between major depression at T-1 and low back pain at T-2
- Line 313: Ligh and oderate drinkers
- References: standardize manuscript titles (are the initial letters capitalized or not? standardize); standardize journal names (abbreviated or in full);
- Line 342: Chronic spinal disorders and psychopathology. research findings (dot)
- Line 380: 2nd edition In Services, D. o. H
Reviewer 4 Report
The paper is readable and clear. Nevertheless, I would recommend thorough linguistic correction (grammar and style) throughout the paper. I've noticed some grammatical errors and other simple mistakes.English should be carefully checked. The paper is well structured and focused on a very interesting topic.
The methodological concept is clear, and the selected methodology is scientifically appropriate. But the its presenatation is very poor. I recommend to increase Study population, Low back pain, Major Depression and Statistical Analysis.
Further, I recommend rewriting the conclusions. The concluding remarks should be more specific and better explained. I miss more future directions and limitations.
In summary, the article is sufficiently interesting to warrant publication, but it needs major revision. Please follow all the comments above.
Round 2
Reviewer 3 Report
Congratulations, accept in present form
Reviewer 4 Report
I agree with all changes made.